# GFI1B and LSD1 repress myeloid traits during megakaryocyte differentiation
Jeron Venhuizen [1,6], Maaike G. J. M. van Bergen [1,6], Saskia M. Bergevoet[1], Daan Gilissen [1], Cornelia G. Spruijt[2], Laura Wingens [3], Emile van den Akker [4], Michiel Vermeulen [2,5], Joop H. Jansen[1], Joost H. A. Martens [2,7] ✉ & Bert A. van der Reijden [1,7] ✉

The transcription factor Growth Factor Independence 1B (GFI1B) recruits Lysine Specific Demethylase 1 A (LSD1/KDM1A) to stimulate gene programs relevant for megakaryocyte and platelet biology. Inherited pathogenic GFI1B variants result in thrombocytopenia and bleeding propensities with varying intensity. Whether these affect similar gene programs is unknow. Here we studied transcriptomic effects of four patient-derived GFI1B variants (GFI1B$^{T174N, H181Y, R184P, Q287*}$) in MEG01 megakaryoblasts. Compared to normal GFI1B, each variant affected different gene programs with GFI1B$^{Q287*}$ uniquely failing to repress myeloid traits. In line with this, single cell RNA-sequencing of induced pluripotent stem cell (iPSC)-derived megakaryocytes revealed a 4.5-fold decrease in the megakaryocyte/myeloid cell ratio in GFI1B$^{Q287*}$ versus normal conditions. Inhibiting the GFI1B-LSD1 interaction with small molecule GSK-LSD1 resulted in activation of myeloid genes in normal iPSC-derived megakaryocytes similar to what was observed for GFI1B$^{Q287*}$ iPSC-derived megakaryocytes. Thus, GFI1B and LSD1 facilitate gene programs relevant for megakaryopoiesis while simultaneously repressing programs that induce myeloid differentiation.

Blood stem cells develop into a variety of hematopoietic cells, including platelet producing megakaryocytes, through a complex process that requires intricate regulation of different transcription factors. The transcription factor Growth Factor Independent 1B (GFI1B) and its paralogue GFI1 are essential for adult hematopoietic stem cells maintenance[1]. GFI1 is expressed throughout different lymphoid differentiation stages and plays an essential role in myeloid development[2,3]. In parallel, GFI1B plays an essential role in the formation of erythrocytes and megakaryocytes[4]. GFI1/1B control gene expression through the recruitment of the chromatin modifying CoREST complex. This is a large protein complex that represses its target genes through deacetylation and demethylation[5,6]. CoREST consists of RCOR and HDAC proteins[6], as well as Lysine Specific Demethylase 1 (LSD1/KDM1A)[7]. LSD1 was first identified as a demethylase, but recent evidence shows that it also acts as a CoREST scaffolding protein in blood cells[4,8–11]. It links GFI1/1B to other CoREST members through interaction with the N-terminal Snail/GFI1

(SNAG) domain of GFI1/1B to form a functional multi-protein complex[9,12]. In turn, GFI1/1B recruit the complex to the DNA through its C-terminal zinc fingers three, four, and five, leading to downregulation of its target genes.

The GFI1B-LSD1 interaction can be disturbed using LSD1 inhibitors like GSK-LSD1[7,9,12–15]. They occupy the catalytic domain of LSD1 which blocks the interaction with GFI1B in hematopoietic cells[10,13,16]. *Lsd1* knockdown in mice impaired differentiation of megakaryocytes, erythrocytes, and granulopoiesis, while monopoiesis was stimulated[17]. Consequently, a LSD1 inhibitor, Bomedemstat, is currently being evaluated in clinical trials to reduce platelet counts in patients with thrombocytosis and myeloproliferative disorders[18]. Inherited variants in GFI1B itself play a role in bleeding disorders with varying intensity. Truncating mutations in the most C-terminal DNA-binding zinc finger cause thrombocytopenia, moderate to severe bleedings, and various megakaryocyte and platelet abnormalities[13,19–22]. In mice, truncating mutations are linked to a milder

[1]Department of Laboratory Medicine, Laboratory of Hematology, Radboud University Medical Center, Research Institute for Medical Innovation, Nijmegen, The Netherlands. [2]Department of Molecular Biology, Faculty of Science, Oncode Institute, Radboud University Nijmegen, Nijmegen, The Netherlands. [3]Department of Molecular Developmental Biology, Faculty of Science, Radboud University Nijmegen, Nijmegen, The Netherlands. [4]Department of Hematopoiesis, Sanquin Research and Landsteiner Laboratory, Amsterdam, Amsterdam, The Netherlands. [5]Division of Molecular Genetics, The Netherlands Cancer Institute, Plesmanlaan 121, Amsterdam, The Netherlands. [6]These authors contributed equally: Jeron Venhuizen, Maaike G. J. M. van Bergen. [7]These authors jointly supervised this work: Joost H. A. Martens, Bert A. van der Reijden. ✉e-mail: joost.martens@ru.nl; Bert.vanderReijden@radboudumc.nl

phenotype and missense mutations in non-DNA binding zinc fingers are linked to milder disease phenotypes in humans[19,23–26].

Currently, it remains unclear how different GFI1B mutations affect the transcriptome in developing megakaryocytes. Here, we elucidated transcriptomic changes caused by one dominant-negative C-terminal truncating mutation (GFI1B^Q287*) and three dominant-negative upstream missense mutations (GFI1B^T174N, GFI1B^H181Y, GFI1B^R184P) in megakaryocytes. Among the missense variants, the GFI1B^T174N variant may not be pathogenic[27]. We observed that each variant has different effects on the megakaryocyte transcriptome. Remarkably, GFI1B^Q287* expression resulted in derepression of myeloid genes in megakaryocytes. Treatment with an LSD1 inhibitor recapitulated these findings. These data indicate that GFI1B and LSD1 act together to repress myeloid genes during megakaryopoiesis.

## Results and discussion
### GFI1B^Q287* fails to repress innate immunity genes in megakaryoblast-like cells

We assessed transcriptomic changes induced by four GFI1B variants linked to bleeding disorders to varying degrees. (Fig. 1). To that extent, we retrovirally expressed GFI1B^T174N, GFI1B^H181Y, GFI1B^R184P, GFI1B^Q287*, wild type GFI1B, or GFP in megakaryoblast MEG01 cells followed by RNA-sequencing. This resulted in comparable (2−3 fold) GFI1B induction in all samples compared to empty-vector transduced MEG01 cells, while LSD1 expression remained stable (Supplementary Fig. 1). Principal component analysis showed all replicates of a sample clustering together. Furthermore, GFI1B and its variants were separated from GFP by PC2, albeit with little variance (Fig. 2). Together, this shows that GFI1B variants are expressed at similar levels across samples and that they have distinct effects on the transcriptome compared to control. Identifying genes that characterize each condition proved difficult due to the number of comparisons needed for classical differential gene expression analysis, since this reduced statistical power considerably. Therefore, we utilized Weighted Gene Co-Expression Network Analysis[28] that merges co-expressed genes into gene modules and correlates them to each condition. This yielded 12 modules containing between 22 and 1005 genes (Supplementary Data 1). 10 modules significantly correlated with at least one condition except for empty vector (GFP) MEG01 cells. They did not show significant correlation with any module (Fig. 3a). Earlier studies showed that alteration of GFI1B by either upregulation or depletion resulted in dysfunctional erythropoiesis[11,29], indicating that GFI1B dosage is important. This is in line with our findings showing that forced expression of any variant or wild type GFI1B induced

transcriptomic changes. We performed non-supervised hierarchical clustering of the module gene expression data to confirm that module correlations in GFI1B variants correspond to higher gene expression in these samples. This grouped all replicates of a condition together and confirmed identified module activities on a per gene basis (Fig. 3b). All tested GFI1B conditions associated with unique sets of modules that converge to different biological pathways. For instance, normal GFI1B caused inhibition of modules 9 and 11 genes, which are implicated in stress response and the immune system, respectively (Fig. 3a, Supplementary Data 2). Module 9 genes were active downstream of GFI1B^H181Y, indicating that this variant failed to repress this gene program. Module 11 genes were suppressed by GFI1B^T174N but activated downstream of GFI1B^R184P and GFI1B^Q287* (Fig. 3a). This suggests that regulation of this module is preserved by the GFI1B^T174N variant but not by GFI1B^R184P or GFI1B^Q287*. Thus, the repression of certain gene programs downstream of normal GFI1B appears to be lost by different GFI1B mutants. The strongest correlation was observed for module 10 that was uniquely active in GFI1B^Q287* replicates (Fig. 3a). This module contained 39 genes related to innate immunity, such as SPI1, GFI1, and CEBPA, that normally stimulate myeloid differentiation[30–32] (Fig. 3c, Supplementary Data 2). These data indicate that only GFI1B^Q287* fails to repress innate immunity genes which may contribute to disturbed megakaryopoiesis and impaired platelet production and function resulting in bleedings in affected cases.

### GFI1B represses myeloid differentiation during megakaryopoiesis

Patients with the GFI1B^Q287* mutation suffer from megakaryocyte abnormalities and bleeding tendencies. To investigate whether GFI1B^Q287* also fails to repress myeloid genes during megakaryopoiesis, we studied the single-cell transcriptome of megakaryocytes differentiated from patient-derived GFI1B^Q287* induced pluripotent stem cells (iPSCs) and compared them to the single-cell transcriptome from wild type megakaryocytes. Single cell transcriptome analysis showed that wild type iPSCs predominantly formed megakaryocytes and megakaryocyte progenitors (75%) and smaller populations of erythroblasts (10%), myeloid and myeloid progenitors (13%), and lymphoid-myeloid progenitors (1%). The megakaryocyte/myeloid ratio was 75%/13% = 5.7. While GFI1B^Q287* iPSCs formed erythroblasts at a similar rate as wild type (7%), the lymphoid-myeloid progenitors population increased to 9% and the megakaryocyte/myeloid cell ratio dropped to 1.3 (Fig. 4a, b). This indicates that GFI1B^Q287* severely affects iPSC-derived megakaryocyte formation. Importantly, differential

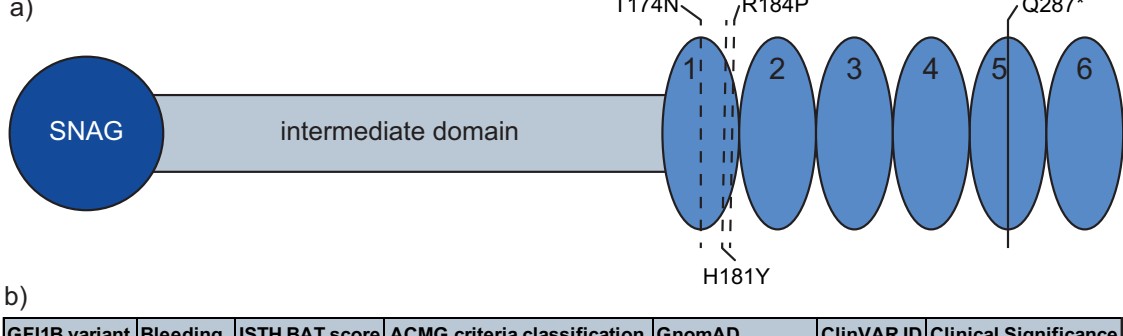

| GFI1B variant | Bleeding | ISTH BAT score | ACMG criteria classification | GnomAD | ClinVAR ID | Clinical Significance |
|---|---|---|---|---|---|---|
| T174N | Unknown | ND | ND | 9-132989071-C-A | 1012632 | Uncertain |
| H181Y | Yes | 4-11 | PM2, PS3-P, PP3, PP1: VUS | ND | ND | ND |
| R184P | Yes | 0-7 | PM2, PS3-P, PP3: VUS | 9-132989101-G-C | 1684454 | Likely pathogenic |
| Q287* | Yes | 7-25 | ND | ND | 102428 | Pathogenic |

**Fig. 1 | Schematic overview of the GFI1B variants investigated. a** The schematic overview for GFI1B includes the N-terminal Snail/GFI1 domain (SNAG), the intermediate domain and the six C-terminal zinc fingers, labeled 1 to 6. In total four GFI1B variants were investigated: Three miss-sense mutations in the first zinc finger and one truncating mutation in zinc finger 5. **b** Clinical data overview for the 4 variants investigated. GFI1B^H181Y, GFI1B^R184P, and GFI1B^Q287* resulted in bleedings with varying intensity. Bleeding tendencies for GFI1B^T174N has not been reported publicly. GFI1B^R184P and GFI1B^Q287* have been reported to be likely pathogenic and pathogenic in ClinVAR, respectively. Information obtained from GnomAD (accessed 19.12.2023)[52], ClinVAR (accessed 19.12.2023), refs. [19,20].

gene expression analysis revealed upregulation of myeloid genes within GFI1B$^{Q287*}$ megakaryocytes compared to wild type megakaryocytes (Fig. 5; Supplementary Data 3). Together, this shows that GFI1B$^{Q287*}$ fails to repress myeloid gene expression which likely contributes to impaired megakaryopoiesis.

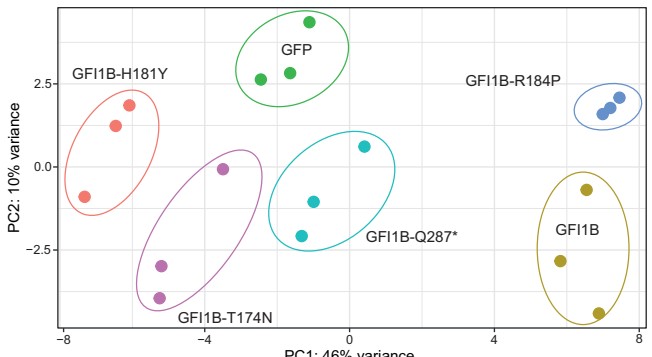

**Fig. 2 | Principal component analysis of MEG01 RNA sequencing data following expression of bleeding-associated GFI1B variants.** RNA sequencing of MEG01 cells with forced retroviral expression of different GFI1B variants, wild type GFI1B and empty vector control (GFP) was done. PCA showed that replicates cluster per sample.

## GFI1B represses myeloid differentiation through SPI1 and other transcription factors

Next, we sought to identify transcription factors downstream of GFI1B that are implicated in repression of myeloid traits using the Single-Cell rEgulatory Network Inference and Clustering (SCENIC) algorithm. SCENIC determines active regulons, which are defined as sets of genes that are regulated as a unit by one transcription factor. As expected, the myeloid-associated SPI1 and ETV5 regulons were active in wild type myeloid cells while the megakaryocytic-erythrocytic GATA1 and TAL1 regulons were active in wild type developing megakaryocytes. Strikingly, SPI1 and ETV5 regulons were not restricted to GFI1B$^{Q287*}$ myeloid cells. Instead, they were active in most GFI1B$^{Q287*}$ cells, including megakaryocytes. This is in line with earlier studies showing that GFI1B suppressed myeloid differentiation of CD34+ cells through SPI1 inhibition[33]. Furthermore, GATA1 and TAL1 regulons showed reduced activity in GFI1B$^{Q287*}$ megakaryocytes (Fig. 4d−i). Our findings indicate that GFI1B facilitates megakaryopoiesis by driving megakaryocyte-specific regulons and repressing myeloid regulons at the same time.

## GFI1B is a major interactor of LSD1 in K562 cells

GFI1B has been found to interact with LSD1 in several different cell types, including MEG01, HEL, and K562 cells[7,12,13,34]. To confirm this interaction, we tagged the *LSD1* gene with GFP at its C-terminus in the *GFI1B* expressing megakaryocyte-erythrocyte progenitor cell line

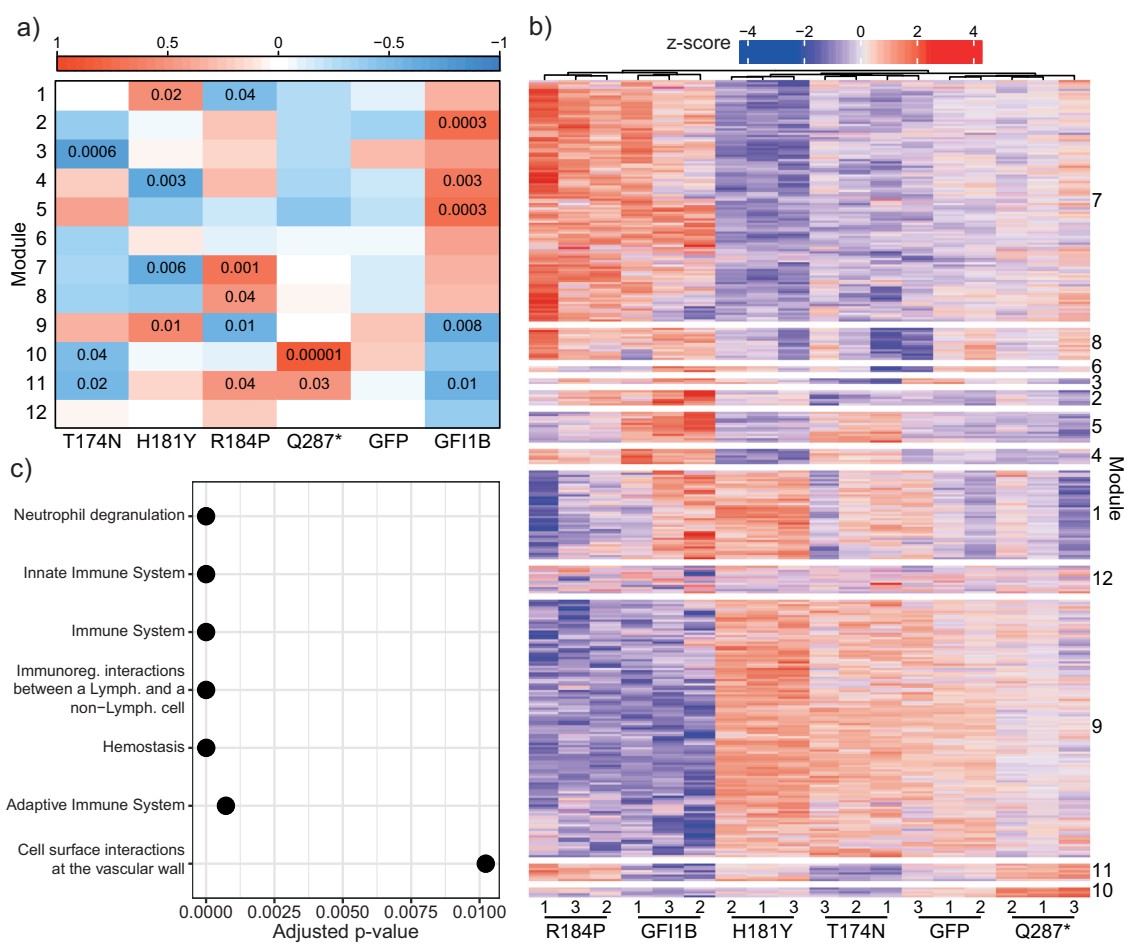

**Fig. 3 | GFI1B variants exert different transcriptomic changes in MEG01 cells. a** Weighted Gene Co-Expression Network Analysis revealed 12 modules, of which 10 modules were significant in at least one condition (see Supplementary Data 1). Module 12 contains all genes that were not be assigned to a module and module 6 did not return a significant correlation with any condition. Every condition had at least one significantly correlating module except for GFP. The strongest and most

significant correlation was found between module 10 and GFI1B$^{Q287*}$. **b** Expression analysis of all module genes showed that high module correlation indeed corresponded to high expression of those genes in that condition. Unsupervised hierarchical clustering grouped all replicates of a condition together. **c** REACTOME enrichment for module 10 using g:Profiler. Most genes in module 10 belonged to pathways that are involved in the innate immune system and homeostasis.

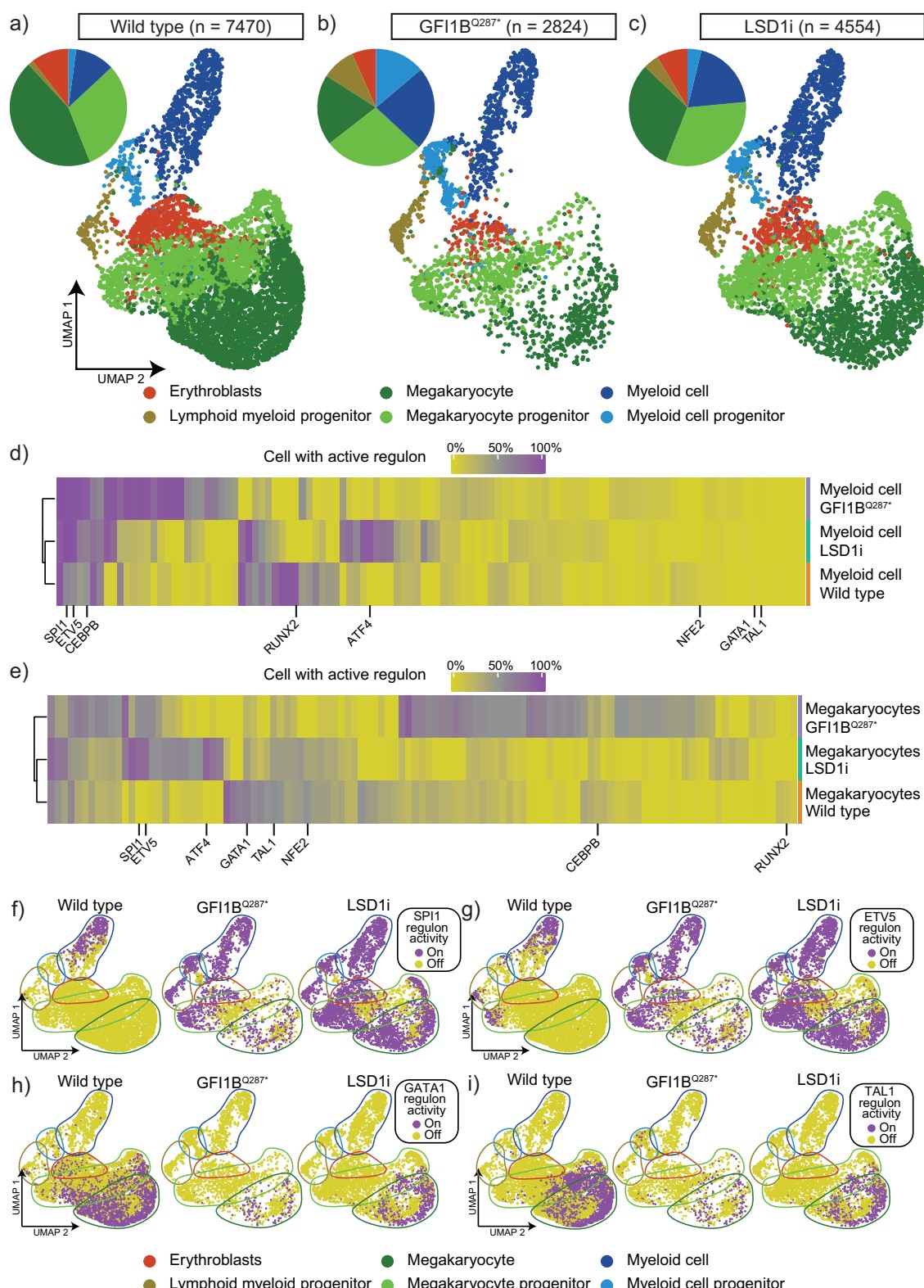

K562. Subsequent GFP immunoprecipitation followed by label-free mass-spectrometry identified 22 interacting proteins, including LSD1, confirming the interaction with established core CoREST members RCOR1/2/3 and HDAC1/2, as well as PHF21A and HMG20A/B. Additionally, it showed interaction with the CtBP complex members ZMYM2/3, RREB1, ZNF217 and CTBP1 (Fig. 6a). It is known that the CoREST complex is part of the CtBP containing complex[35–37],

which our data confirms. LSD1 interacts with the SNAG domain of transcription factors and our mass-spectrometry analysis revealed interaction with the SNAG-domain containing transcription factor GFI1B (Fig. 6a). Because the interaction between GFI1B and LSD1 has been described in detail in hematopoietic systems, we wondered if LSD1 inhibition resulted in a similar impaired megakaryocyte development as GFI1B[Q287*].

**Fig. 4 | scRNA sequencing reveals a myeloid differentiation bias by GFI1B^Q287* or GSK-LSD1 treatment (LSD1i) during megakaryocyte differentiation of iPSCs.** UMAP showing iPSC-derived wild type (**a**), GFI1B^Q287* (**b**), and GSK-LSD1 inhibitor treated (**c**) cells after megakaryocyte differentiation. Cell type annotation revealed erythroblasts (red), megakaryocytes (dark green), megakaryocyte progenitors (light green), myeloid cell (dark blue), myeloid cell progenitors (light green), and lymphoid-myeloid progenitors (gold). The megakaryocytes and megakaryocyte progenitors to myeloid and myeloid progenitor cell ratio was 5.7, 1.3 and 2.7 for the wild type, GFI1B^Q287* and LSD1i conditions, respectively. **d** Heatmap showing the activity of gene programs that are regulated by a transcription factor (regulons). SPI1, ETV5 and CEBPA were active in myeloid cells in all conditions. Regulons controlled by megakaryocytic transcription factors such as GATA1, TAL1, and NFE2 were not active in myeloid cells of any condition. **e** The GATA1, TAL1, and NFE2 regulons were active in 50% of all wild type megakaryocytes or more.

NFE2 was not found to be active in GFI1B^Q287* megakaryocytes and GATA1 and TAL1 were active in 22% and 29% of GFI1B^Q287* megakaryocytes, respectively. Myeloid SPI1 and ETV5 were not active in wild type megakaryocytes but showed activity in LSD1i and GFI1B^Q287* megakaryocytes. **f** Regulon activity was projected onto our UMAP data. The myeloid-related SPI1 regulon was active in myeloid cells in the wild type condition and in all cell types in the GFI1B^Q287* and LSD1i conditions including megakaryocytes. **g** SCENIC revealed that the ETV5 regulon was only active in myeloid cells in the wild type condition. In contrast, it was active in all cell types in the GFI1B^Q287* and LSD1 inhibitor-treated condition. **h** The erythro-megakaryocytic GATA1 regulon was active in wild type megakaryocytes. Its activity was reduced in GFI1B^Q287* and LSD1i megakaryocytes. **i** SCENIC revealed that TAL1 was active in wild type megakaryocytes. The activity of the TAL1 regulon was reduced in LSD1 inhibitor-treated megakaryocytes and almost absent in GFI1B^Q287* megakaryocytes.

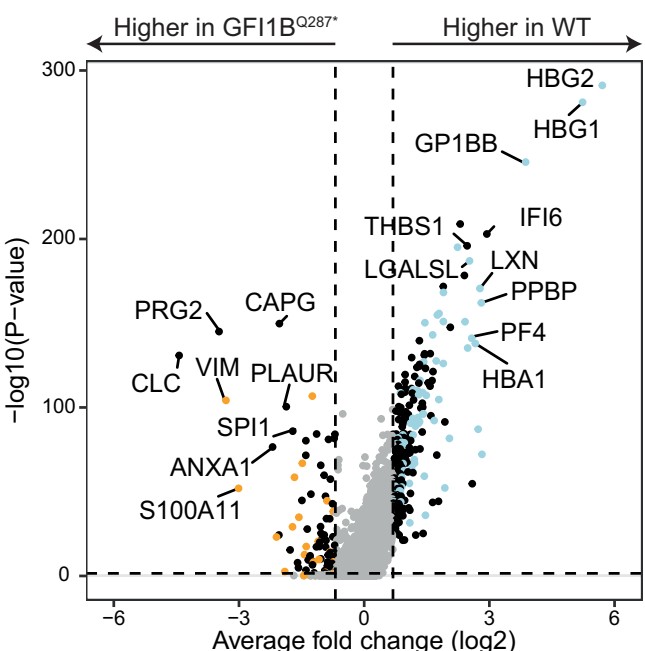

**Fig. 5 | Differential gene expression analysis of wild type and GFI1B^Q287* megakaryocytes derived from iPSCs revealed upregulation of myeloid genes in the GFI1B^Q287* megakaryocytes.** Platelet-related genes were upregulated in the wild type sample. Genes that are also differentially expressed when comparing wild type and GFI1B^Q287* megakaryocytes are annotated in orange (also upregulated in GSK-LSD1 inhibitor (LSD1i) treated megakaryocytes) and blue (also upregulated in wild type megakaryocytes compared to LSD1i megakaryocytes). For full list see Supplementary Data 3.

### GFI1B and LSD1 repress similar myeloid traits during megakaryocyte development

We inhibited the GFI1B-LSD1 interaction using GSK-LSD1[13] to determine whether this interaction is required to inhibit myeloid traits in megakaryocytes. To this end, MEG01 cells were exposed to GSK-LSD1 for 72 h, followed by bulk RNA-sequencing. Analysis of module 10 innate immunity genes that were identified in GFI1B^Q287* expressing MEG01 cells showed that the vast majority were also active following LSD1 inhibition (Fig. 6b). This indicates that the GFI1B-LSD1 interaction is required to suppress these genes in MEG01 cells. Next, we investigated whether LSD1 inhibition also fails to repress myeloid traits during iPSC-derived megakaryopoiesis. To this end, wild type iPSCs were differentiated towards megakaryocytes and treated with GSK-LSD1 for only 24 h prior to single-cell RNA-sequencing. Similar to GFI1B^Q287* iPSC-derived megakaryopoiesis, the megakaryocyte/myeloid cell ratio dropped from 5.7 in wild type cells to 2.7 after LSD1 inhibition (Fig. 4a, c). Additionally, myeloid genes were upregulated in LSD1-inhibitor treated megakaryocytes compared to wild type

megakaryocytes, some of which were also upregulated in GFI1B^Q287* megakaryocytes (Figs. 5, 6c; Supplementary Data 3). Prolonged treatment of wild type iPSC-derived cells with GSK-LSD1 revealed the absence of the megakaryocyte marker CD42b and an increase of the myeloid marker CD86 (Fig. 6d). Thus, the GFI1B-LSD1 interaction is fundamental to megakaryocyte development. Finally, we determined which regulons were activated upon LSD1 inhibition using single-cell RNA sequencing of iPSC-derived megakaryocytes. Again, SPI1 and ETV5 regulons were found to be active in all cell types including megakaryocytes. GATA1 and TAL1 regulon activities were reduced compared to non-treated cells, comparable to the effects of GFIB^Q287* (Fig. 4d−i). Additionally, some regulons were affected by LSD1 inhibition but not the GFI1B^Q287* mutation, such as ATF4 (Fig. 4d, e). Recently, it was shown that ATF4 is downregulated upon LSD1 inhibition potentially through a loss of the CREBBP-LSD1 interaction in glioblastoma cells[38]. In a lung carcinoma cell line, ATF4 was downregulated through increased H3K9me2 levels after LSD1 inhibition[39]. On the contrary, we found induction of ATF4 regulon activity upon LSD1 inhibition, suggesting that the mechanisms of action following LSD1 inhibition are different in our model system. More detailed studies that determine early chromatin binding, epigenetic, and transcriptional changes following GFI1B^Q287* expression and LSD1 inhibition may reveal which activities are at play during megakaryopoiesis.

Summarizing, our data demonstrate that different inherited GFI1B mutations affect the megakaryocyte gene programs in unique ways. The most severe bleedings in patients caused by the GFI1B^Q287* mutation could be explained by the loss of GFI1B-LSD1 mediated suppression of myeloid traits during megakaryopoiesis, potentially through derepression of SPI1 and ETV5. SPI1 is a direct GFI1B target and loss of GFI1B-mediated SPI1 suppression has been linked to myelomonocytic differentiation[33]. LSD1 has been found to be required for terminal differentiation of various hematopoietic cell types except monocytes[17]. Our own data confirms and expands on these findings. We propose that GFI1B and LSD1 work together to suppress SPI1 and other myeloid transcription factor programs during megakaryopoiesis. Loss of GFI1B or LSD1 function deregulates these programs inducing myeloid differentiation. This could explain perturbed megakaryocyte development and platelet production in GFI1B^Q287* patients[20] and reduced platelet formation in patients treated with the LSD1 inhibitor Bomedemstat[18].

## Methods
### MEG01 RNA sequencing and network analysis
MEG01 cells were cultured at 5% CO$_2$ and 37 °C in RPMI 1640 (GIBCO) supplemented with 10% heat-inactivated fetal calf serum. Cells were transduced using pMIGR1-GFI1B_variant-IRES-GFP constructs and FACS-sorted for GFP (BD FACSAria). RNA was isolated using the NucleoSpin RNA plus kit (Macherey-Nagel). Library generation was performed on 100 ng RNA using the KAPA RNA HyperPrep Kit with RiboErase (HMR;Kapa Biosystems), with a RNA fragmentation of approximately 300 bp fragments for 6 min at 94 °C. Library size distribution was measured using High Sensitivity DNA analysis on an Agilent 2100

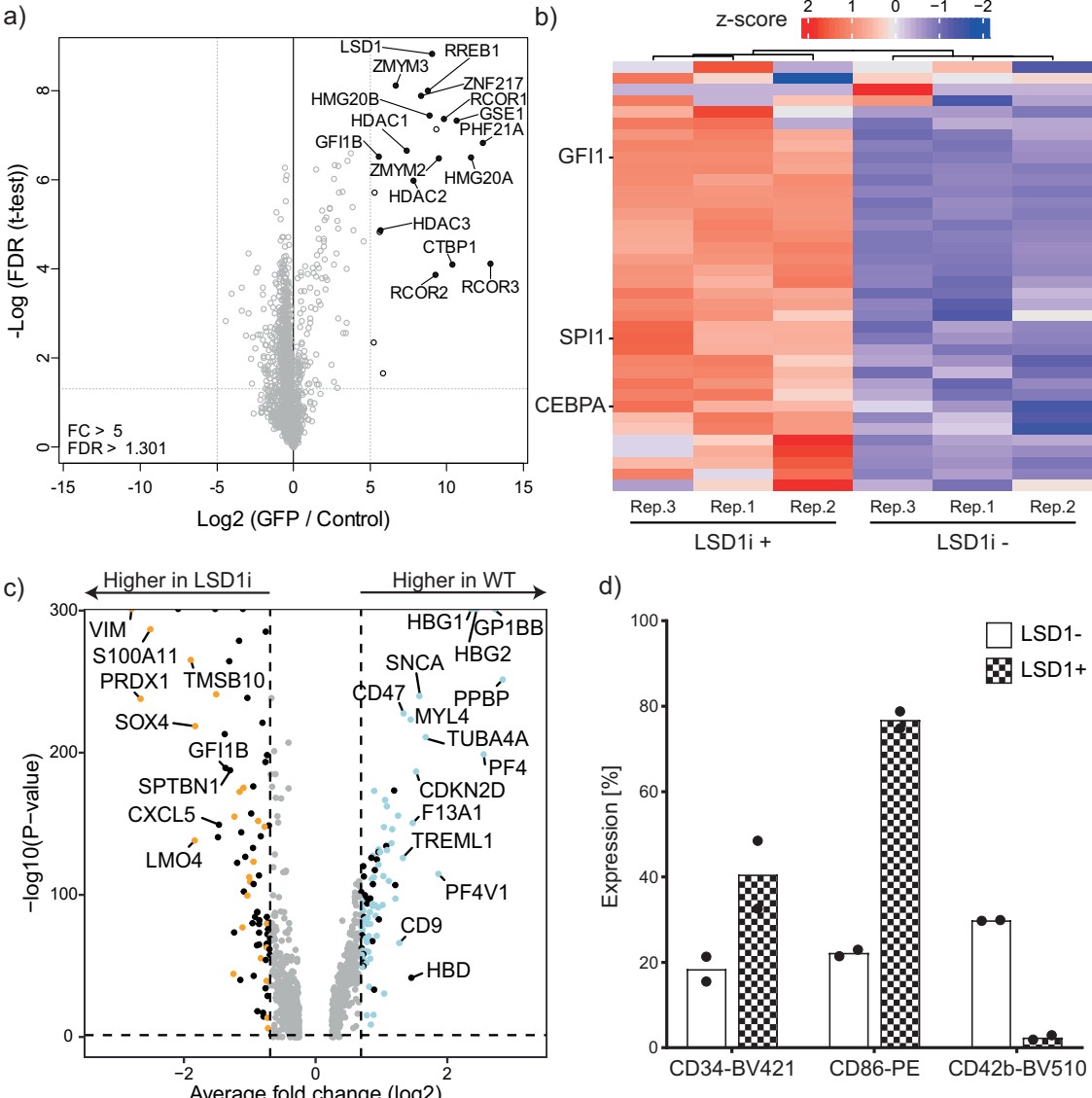

**Fig. 6 | LSD1 and GFI1B repress myeloid traits in megakaryocytes derived from iPSCs. a** The endogenous *LSD1* gene was tagged with GFP in K562 cells and mass-spectrometry was performed after GFP pulldown. We identified LSD1-interacting proteins after pulldown, showing that the CoREST members RCOR1/2/3, HDAC1/2, PHF21A, HMG20A/B, and GFI1B interacted with LSD1. Additionally, members of the CtBP complex were found to interact with LSD1 (solid dots). **b** MEG01 cells were treated with the small molecule GSK-LSD1 inhibitor and RNA sequenced. Module 10 genes that were identified to be highly expressed in GFI1B^Q287* MEG01 cells were also highly expressed after LSD1 inhibition. **c** Differential gene

expression of ISPC-derived scRNA-seq data showed that LSD1 inhibition leads to the upregulation of myeloid genes in megakaryocytes. Genes that were also upregulated in GFI1B^Q287* megakaryocytes are highlighted in orange, while genes also upregulated in wild type megakaryocytes are highlighted in blue. **d** Wild type iPSCs were differentiated to megakaryocytes and treated with an LSD1 inhibitor for 72 h before harvesting. FACS analysis showed that LSD1 inhibition resulted in induced expression of CD86 and CD34 in megakaryocytes compared to untreated wild type cells. CD42b expression was absent in treated cells compared to untreated ($n = 2$ independent experiments, bars denote mean, points are individual data points).

Bioanalyzer (Agilent) and its corresponding software (version B.02.08.SI648). Libraries with average sizes between 300 and 400 bp were sequenced on the NextSeq 500 (Illumina). Data was quality-controlled, mapped, and count matrices were generated using nextflow pipelines. Count matrices were loaded into R (v.4.1.1), and variance-stabilized counts were generated using DESeq2 (v1.32). The top 5% most variable genes were used as input for Weighted Gene Co-expression Network analysis[28] (v1.71). The softpower threshold was set to 12 and the networks were calculated using signed-hybrid networks. Resulting modules were merged using a cutting height of 0.2. All module genes were uploaded to gProfiler[40] (version e109_eg56_p17_1d3191d, accessed 04.08.2023). REACTOME enrichment was performed as a multi-query using all expressed genes (cumulative count per gene > 10 and expressed in at least 50% of all samples) as a statistical domain scope.

MEG01 and induced pluripotent stem cells were treated with 4 μM of the GSK-LSD1 inhibitor (Merck) for 24 h, 48 h or 4 days, as indicated.

### Induced pluripotent stem cell maintenance and differentiation towards megakaryocytes
Induced pluripotent stem cells were maintained according to previously described protocols on Matrigel (Corning), E8 medium (Thermofisher Scientific), and mTESR1 medium (Stemcell Technologies)[41,42]. The medium was regularly changed and cells were passaged with TrypLE Select (Thermofisher Scientific). One day prior to differentiation, the cells were clump passaged using ReLeSR (Stemcell Technologies). Cells were differentiated towards hematopoietic stem cells using the STEMdiff Hematopoietic kit (Stemcell technologies). From day 10 onwards, hematopoietic stem cells were harvested in modified Iscove modified Dulbecco medium

(HEMAdef)[43] according to protocol and to stimulate megakaryocyte differentiation supplemented with a cytokine cocktail of 10 ng/ml rhIL-3, 10 ng/ml rhIL-6, 10 ng/ml rhIL-9, 10 ng/ml rhIL-11, 10 ng/ml VEGF, 10 ng/ml BMP4, 1% Stem Cell Factor (Immunotools), 50 ng/ml Thrombopoietin (Peprotech), and 1% Insulin Transferrin Selenium (Thermofisher Scientific)[13,44].

### Cell analysis by flow cytometry

Flow cytometric analyses were performed on a Gallios flow cytometer (Beckman Coulter) and analyzed using Kaluza software v.2.1.2 (Beckman Coulter). Differentiated iPSCs were stained using the surface marker antibodies CD34-561 Brilliant Violet 421 (Biolegend), CD86-IT2.2 PE (Biolegend), and CD42b-HIP1 Brilliant Violet 510 (Biolegend). Cells were gated for live cells based on forward/side scatter area and singlets were gated based on forward scatter area/side scatter area (Supplementary Fig. 2) For single cell RNA sequencing, living cells were sorted using 7AAD (Sigma) as a viability marker on the FACS Aria (BD biosciences) using a 100 μm nozzle (BD Biosciences).

### Single cell RNA sequencing data processing and analysis

Single-cell RNA-sequencing was performed using the Chromium Single Cell 3' kit (v3.1 Chemistry; 10X genomics) according to protocol. Sorted cells were loaded at a density of ~1200 cells/μl with a desired recovery of 10.000 cells per sample. Quality and size of the generated libraries were verified using the Qubit 1x dsDNA HS assay kit (Invitrogen) and the Bioanalyzer (Agilent). Sequencing was performed on the Nextseq500 (Illumina) at an average sequencing depth of ~266 million reads. Raw data were processed using Cellranger v.6.0.0 (10x Genomics) and aligned against the GRCh38 transcriptome, and feature-barcoded matrices were created. Empty droplets were filtered using the emptydrops function from the DropletUtils package (v1.10.3)[45] using 30,000 iterations and FDR of 0.01 and 100 UMIs as the lower bound UMI threshold for non-empty droplets. The Seurat package (v.4.0.1)[46] was used for downstream analysis and quality checks within the R environment (v.4.0). Dimensional reduction was performed on retained droplets, and a shared nearest neighbor graph was constructed to identify clusters using Louvain clustering. Doublets were filtered using the DoubletFinder package (v.2.0.3)[47]. Clusters were identified using the shared nearest neighbor (SNN) approach and selecting the Louvain clustering method. Clustree (v.0.4.1)[48] was used to identify the optimal clustering resolution. High mitochondrial gene content indicates that cells were damaged during sample preparation, because cytoplasmic RNA dissipates from the damaged cell while mitochondria are retained. Clusters with a high proportion of mitochondrial genes and no megakaryocyte marker expression were removed from the analysis (cluster 6 and 19 for LSD1i sample, cluster 14, 16, 23 for GFI1B$^{Q287*}$ and none for wild type). After filtering, 7470 wild type cells, 4554 GSK-LSD1-treated (LSD1i) cells and 2824 GFI1B$^{Q287*}$ cells passed quality control (Supplementary Fig. 3). Samples were normalized using SCTransform and integrated using the FindIntegrationAnchors function. Cell cycle was assessed using CellCycleScoring. Dimensional reduction using Uniform Manifold Approximation and Projection (UMAP) showed that cells were clustering according to their cell cycle which is a confounder in our analysis (Supplementary Fig. 3f). Therefore, we regressed out cell cycle during sample integration to exclude any effect based on cell cycle. Following the cell cycle regression, the samples were re-integrated based on the FindIntegrationAnchors function. Clusters were identified using Louvain clustering, and resolution 0.3 was the optimal cluster separation (Supplementary Fig. 4a, b). Clusters were annotated using two independent annotation tools. First, we performed single cell gene set enrichment analysis with the escape package (v.1.0.1). For single cell gene set enrichment analysis, the C8 cell type signature gene set was downloaded from the MSigDB website (Database v7.4). Secondly, we used Azimuth in combination with the publicly available Azimuth PBMC data set[46,49]. We merged both annotations obtained by each tool, manually renamed each cluster and determined to top expressed genes per cluster (Supplementary Fig. 4c–e). Differential

expressed genes were identified using the FindMarkers function with a minimal expression percentage of 25 in at least one cluster. Genes were considered differentially expressed with log2 fold change <−0.69 or >0.69 and adjusted $p < 0.05$.

### SCENIC analysis

We calculated the gene regulatory network for each sample independently, using the python SCENIC implementation and Arboreto allowing for multi-processing[50]. We ran the pipeline 100 times for each sample to account for random number generation. The iteration outputs were merged per sample, and regulons (i.e., transcription factors and its associated target genes) that were found in at least 95% of the iterations were retained. We calculated the regulon activity using AUCell[50]. Activity scores were binarized for each sample to annotate the On/Off state for every regulon in a sample. This enabled between sample comparison. AUC thresholds for regulon binarization were calculated automatically, and curve fit was checked manually as described by the authors of (py)SCENIC. Binary regulon score for each regulon in each cell are supplied as Supplementary Data 4.

### CRISpainting LSD1 and mass-spectrometry

K562 cells were GFP CRISPainted at the endogenous LSD1 locus according to Schmid-Burgks protocol[51]. $680 \times 10^6$ K562 cells with endogenously tagged LSD1 were harvested for mass-spectrometry. Nuclei were isolated and GFP pulldown was performed using GFP nano-trap beads (Chromotek). Beads were combined with 2 mg nuclear extract in buffer C (300 mM NaCl, 20 mM Hepes KOH pH 7.9, 20% v/v glycerol, 2 mM MgCl2, 0.2 mM EDTA, 0.1% NP40, complete protease inhibitors w/o EDTA, 0.5 mM DTT) + 50 μg/ml EtBr and incubated (90 min at 4 degrees). Then, beads were washed two times with buffer C, two times with PBS + 0.25% NP40 and two times with PBS. Liquid was removed and resuspended in elution buffer (2 M Urea in 100 mM Tris pH 8 and 10 mM DTT) at RT and incubated for 20 minutes while shaking. Proteins were digested on beads with trypsin (2 M urea, 100 mM Tris/HCl pH8.5, 10 mM DTT, 0.25 μg Protease). Digested protein were then STAGE tipped. Peptides were measured using a reverse phase column connected to an Easy-nLC1000 (ThermoFisher) coupled to a Thermo Orbitrap Exploris 480 (ThermoFisher Scientific). Data were analyzed using MaxQuant 1.6.6.0.

### Statistics and reproducibility

Statistical analysis for WGCNA and SCENIC has been built into the model as part of the standard workflow. RNA seq samples were done in triplicate based on one biological sample. scRNA analysis and mass-spectrometry was done based one biological sample. Flow data was analyzed based on two biological samples.

### Reporting summary

Further information on research design is available in the Nature Portfolio Reporting Summary linked to this article.

## Data availability

All sequencing data have been submitted to GEO under accession numbers GSE244609 and GSE244756. The mass spectrometry proteomics data have been deposited to the ProteomeXchange Consortium via the PRIDE partner repository with the dataset identifier PXD050401.

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

## Acknowledgements

This work was supported by the Landsteiner Foundation for Blood Transfusion Research (project 1531) and the Radboud Research Institute for Medical Innovation. The Vermeulen lab is part of the Oncode Institute, which is partly funded by the Dutch Cancer Society (KWF).

## Author contributions

M.G.J.M.V.B., J.V., J.H.A.M. and B.A.V.D.R. designed the research project. M.G.J.M.V.B., J.V. and S.B. performed the experiments and analyzed the data. L.W. performed the (sc)RNA sequencing and D.G. ran the bioinformatic pipelines and provided expert knowledge for the bioinformatic analysis. C.G.S. ran the mass-spectrometry experiments and analyzed the raw data under supervision of M.V. E.V.D.A. provided the iPSC cell lines and expert knowledge for culturing and differentiation. J.H.J. critically evaluated the results and research progression. The manuscript was written by J.V., M.G.J.M.V.B. and B.A.V.D.R. and was critically evaluated by all authors.

## Competing interests

The authors declare no competing interest.
