## [Peer Review File · Communications Biology]

Reviewers' comments:

Reviewer #1 (Remarks to the Author):

Venhuizen and colleagues examine the effect of different mutated variants of Gfi1b on the development of megakaryocyte. They conclusively show that the axis of Gfi1b-LSD1 explains the phenotype.

The novelty of this data is limited but still represent a progress towards understanding it. The experiments are conclusively executed from a group with high reputation in the field of Gfi1b.

However in the discussion and introduction part it has some weaknesses. The authors heavily rely of self reference. The Gfi1b field is not as big as other fields and reference of the group of Moroy to Gfi1b and Gfi1 as well as the work of group of Goethert in Gfi1b and LSD1 as well as Enver group describing Gfi1b mutations are omitted in addition to other groups. Fair credit should be give to other groups as well.

The discussion also gets rather floppy with claim that LSD1 are used in thrombocythaemia and that targeting the GFI1B-LSD1-SPI1/ETV5 would represent a possible target to treat patients with Gfi1b mutations.

I would ask the authors to discuss this with a physician experienced in hematology and treatment of malignant diseases.

Reviewer #2 (Remarks to the Author):

The van der Reijden laboratory has published several seminal work on the mutations of the GFI1b gene associated with congenital and acquired disorders of the megakaryocyte cell lineage and how their effects may be therapeutically targeted by inhibitors of the obligatory GFI1b partner LSD1.

The paper currently under review by the van der Reijden laboratory presents the results of gain-of-function experiments in surrogate megakaryocyte models aiming to generate mechanistic insights that would explain the variegation of the traits induced by the different GFI1b mutations in humans. The mutations investigated were the point mutation GFI1BT174N, H181Y and R184P and the missense mutation Q287*. The surrogate models investigated are the MEG01 megakaryoblast cell line and induced pluripotent stem cells derived from a patient carrying the Q287* mutation induced to differentiate along the megakaryocyte lineage. The effects of the different mutations were profiled by single cell RNA seq. Loss-of-function studies were performed by treating the cells with an inhibitor of LSD1.

The most important take-home message of the data presented is that the Q287* mutation appears to encode a dominant negative protein because it induces a molecular signature similar to that induced by the LSD1 inhibitor. This conclusion is clinically important because the phenotype expressed by patients carrying the Q287* mutation may predict the long-term effects of LSD1 treatment in patients with essential thrombocythemia and other myeloproliferative disorders.

1) Although the power conferred to the paper by single cell RNAseq analyses is greatly appreciated, the interpretation of the findings is greatly affected by the paucity of controls. The megakaryoblastic MEG-01

cell line was isolated long time ago (in 1983) from the marrow of a 55-year-old, male patient with Chronic Myelogenous Leukemia. At that time, this cell line contained the Ph chromosome and was hyper-diploid with a modal number of 56-58 chromosomes. The levels of endogenous GFI1b and LDS1 expressed by the cell line at baseline and after transfection should be presented. In addition, some type of assurance that the infections achieve comparable levels of expression of the different GFI1b variants should be presented. This assurance is important because the experiments with over-expression of GFI1b WT suggest that expression levels are important. Presently, it is difficult to assess whether the different phenotypes are sustained by mutation-specific effects or by technical issues which affected the expression levels of the various mutants.

2) Figure 1. The principal component analyses shown in this Figure is confusing. The cluster of the cells transfected with an empty vector (GFP) is separated by that of the cells transfected with GFI1B-q287* and with GFI1B WT. Moreover the GFI1B-Q287* cells resemble more closely GFI1B-T174N cells than WT cells. In any case, the variance is very low, indicating that the cells are overall very similar.

3) Figure 2. Again the module of the GFP cells is very different from that of the GFI1B WT, indicating that dosage is important.

4) Figure 1. The strategy to use mitochondrial genes and cell cycle genes to clean possible confounding factors is unclear given the number of reports indicating that mitochondrial and cell cycle genes are part of the megakaryocyte differentiation signature. Please clarify the rationale supporting this strategy.

5) Supplementary Figure 2 is confusing because the pie charts of the clusters in D do not recapitulate the heat maps in C. The pie chart of cluster 0 indicate a high percentage of megakaryocyte genes which are not present in its heat map. Cluster 0 and 8 appear very similar and it is confusing whether they represent progenitor or endothelial cells. What the word progenitor means, hematopoietic or endothelial progenitor cells? Cluster 4 appears to have a megakaryocyte rather than an Ery signature, as indicated.

6) Page 9: "GFI1BQ287* fails to repress innate immunity genes in megakaryoblast-like cells." Recent data indicate that megakaryocytes represent four different sub-populations each one with a different function (Pariser et al. JCI 2021;131(1),e137377; Yeung et al Blood Adv 2020;4(24),6204-17; Wang et al. Cell Stem Cell 2021; 28(3),535-549.e8; Sun et al. Blood 2021;138(14),1211-24, among others). One of these populations has the signature of immune cells and is involved in the host response toward opportunistic infections in the lung. The expression signature of the different megakaryocyte subpopulations is published. It seems here a missed opportunity that the expression data presented in this manuscript are not compared with the signatures of the megakaryocyte subpopulations already published to assess whether the myeloid effects of these genes are exerted on "myeloid cells" or on "megakaryocyte subpopulations with myeloid functions".

7) Page 10, line 184. What was the rationale to study iPSC carrying the GFI1BQ287* mutation in the context of this study? These experiments should include iPSC carrying the WT mutation as control.

8) Page 11, lines 210-211. The sophistication of the method used for GFP-painting of the endogenous LSD1 gene is greatly appreciated. It is unclear, however, what was the rationale for using the erythroleukemic K562 cell line for these experiments. As the megakaryoblastic MEG-01 cell line, the K562 cell line harbor the Ph chromosome but acquires megakaryocytic features only when treated with PMA. Results on PMA induced K562 cells would have been more relevant but are not presented. What is the copy number of the LDS1 gene in K562 and MEG-01 cells? Was on the basis of this copy number that

determined the cell line to be used in these experiments. In any case, since LDS1 inhibitors are not considered for treatment in chronic myelogenous leukemia but in myeloproliferative disorders, the use of the HEL cell line which carries the JAK2V617F mutation would have provided more clinically relevant information.

9) A diagram summarizing the location of the mutations analyzed, the trait expressed by the patients and the expression signature induced by their over-expression would be greatly beneficial. Ideally, this diagram should also indicate whether the patients carrying these mutations have defective innate immune responses.

10) The title is too restrictive and does not give sufficient credit to the novelty of the message provided by the paper.

11) The conclusions are too dry and do not provide sufficient credit of the reasons why the study is important.

Minor:

- Page 3, lines 48-50 and Ref. 14. Ref. 14 is a non-appropriate self-quotation. It is a review on the role played by LDS1 in AML and does not discuss the use of Bomedemstat to treat thrombocytosis in patients with myeloproliferative disorders, as stated by this paragraph.

- The legend of all the Figures and Tables included in the supplementary should specify if the data refer to the MEG-01 cell line or iPSC derived cells.

- Does the manuscript have technical or conceptual flaws that should prohibit its publication? If so, please provide details.

- Although the sophistication of the technology is greatly appreciated, there are several methodological aspects which were not sufficiently addressed. It is unclear:

- The rationale that guided the choice of the surrogate models, including the patient-derived iPSC cell line instead than WT iPSC.

- Control experiments indicating that the transfections sustained expression of the different mutants at comparable levels. The data on the over-expression of the WT gene do indeed indicate that expression levels are significant.

- The rationale for purging the single cell data of mitochondrial and cell cycle genes. Mitochondrial and cell cycle genes are part of the megakaryocyte phenotype. The statistical analyses of the single cell RNAseq appears too much polished.

- Are the conclusions original? If not, please provide relevant references.

The major problem with the conclusion is that the data, as presented, do not allow to discriminate whether GF11b/LDS1 interaction induce a switch between megakaryocyte and myeloid differentiation or regulate instead the differentiation switch among different megakaryocyte subpopulations. Recent data indicate that megakaryocytes represent four different sub-populations each one with a different function (Pariser et al. JCI 2021;131(1),e137377; Yeung et al Blood Adv 2020;4(24),6204-17; Wang et al. Cell Stem Cell 2021; 28(3),535-549.e8; Sun et al. Blood 2021;138(14),1211-24, among others. One of these subpopulations has a signature of immune cells and is involved in the host response toward opportunistic infections in the lung. The differentiation of immune-megakaryocytes is sustained, as the

signature of the cells discussed by this paper, by SP1. Since the expression signature of the different megakaryocyte subpopulations is published, it seems here a missed opportunity that the expression data presented in this manuscript are not compared with that of the megakaryocyte subpopulations already published to assess whether the myeloid effects of GFI1b/LSD1 are exerted on “myeloid cells” or on “megakaryocyte subpopulations with myeloid functions”.

- Do you feel that the results presented are of immediate relevance for people in your own discipline or for a broader audience? If you recommend publication, please outline briefly what you consider to be the outstanding features.

The results presented have enormous scientific and clinical potential.

At the scientific level, it may clarify how mutations in the GFI1b gene affects cell fate decision at the level of megakaryocyte subpopulations.

At the clinical level, since LSD1 inhibitors are moving to clinical trial, these data have enormous potential relevance. In fact, a careful phenotypic and molecular analyses of the traits induced by GFI1b mutations may provide information to predict long term effects of LSD1 inhibitor-based therapies.

Reviewer #3 (Remarks to the Author):

The manuscript presents genomic data from cells transduced with expression vectors for GFI1B mutants found in human patients with thrombocytopenia and data sets from human ISPCs from patients with a GFI1B mutation. The data are well presented and support the conclusions of the authors.

The manuscript does not have any technical or conceptual flaws and the the conclusions are original. Data that were generated in murine models with Gfi1b mutant alleles to address thrombocytopenia have been published and the results could be discussed in relation to the insight gained here, if there is space.

The results presented are most certainly of immediate relevance for people in this field and even for a broader audience. The most compelling feature of this article is that fact that primary human stem cells have been used and patient derived mutants have been studied. Also hat convergence of results with mutant GFI1B forms and the LSD1 inhibitor nicely corroborates mechanistic models for human patients that have been suggested previously in cell lines or mouse models.

No additional experiments are needed, but the following point could be addressed by text to make a few points clearer:

- The authors use an LSD1 inhibitor, which would not only affect the function of GFI1B but also other transcription factors that associate with LSD1; which ones are those and could that affect the interpretation of the results?
- It has been shown that LSD1 inhibitors lead to the dissociation of LSD1 and GFI1B and to the eviction of both from the chromatin. Is it expected that this happen here as well with the specific inhibitor used

here and the GFI1B mutant forms?

- Also, does LSD1 act as an enzyme or a scaffold to recruit other regulators in this context? The authors mention this shortly, but could expand on that issue if there is space.
- The wild type endogenous GFI1B is still expressed in all the systems that the authors use: is the model then that the mutant forms act in a dominant negative manner? Or is the auto repression of GFI1B affected?
- The introduction should mention that GFI1B has a paralogue (GFI1) and its role in myeloid cells. Also, that GFI1 is most certainly expressed in the stem cells that were used to generate megakaryocytes.

Reviewer #1 (Remarks to the Author):

Venhuizen and colleagues examine the effect of different mutated variants of Gfi1b on the development of megakaryocyte. They conclusively show that the axis of Gfi1b-LSD1 explains the phenotype.

The novelty of this data is limited but still represent a progress towards understanding it. The experiments are conclusively executed from a group with high reputation in the field of Gfi1b.

However in the discussion and introduction part it has some weaknesses. The authors heavily rely of self reference. The Gfi1b field is not as big as other fields and reference of the group of Moroy to Gfi1b and Gfi1 as well as the work of group of Goethert in Gfi1b and LSD1 as well as Enver group describing Gfi1b mutations are omitted in addition to other groups. Fair credit should be give to other groups as well.

The discussion also gets rather floppy with claim that LSD1 are used in thrombocythaemia and that targeting the GFI1B-LSD1-SPI1/ETV5 would represent a possible target to treat patients with Gfi1b mutations.

I would ask the authors to discuss this with a physician experienced in hematology and treatment of malignant diseases.

We thank reviewer 1 for the comments provided. We agree that the work of the Moroy lab, Goethert lab, Enver lab, and others should have been presented in more detail. Therefore, we adapted the introduction and discussion to include their work. Specifically, we included work from the Moroy lab regarding Gfi1 in mouse and the role of GFI1B/GFI1 in hematopoietic differentiation (page 3, ll. 39-43), the work of the Enver lab linking GFI1B to SPI1 in AML (page 11, ll. 231-233 & page 13 ll. 282-284) and the work of the Goethert lab regarding LSD1 knockdown in mice (page 3, ll. 55-56 & page 13 ll. 284-285).

We also adapted the final paragraph of the results/discussion to improve the quality of the discussion (page 12f, ll. 268-278 & page 13 ll. 282-290).

Reviewer #2 (Remarks to the Author):

The van der Reijden laboratory has published several seminal work on the mutations of the GFI1b gene associated with congenital and acquired disorders of the megakaryocyte cell lineage and how their effects may be therapeutically targeted by inhibitors of the obligatory GFI1b partner LSD1.

The paper currently under review by the van der Reijden laboratory presents the results of gain-of-function experiments in surrogate megakaryocyte models aiming to generate mechanistic insights that would explain the variegation of the traits induced by the different GFI1b mutations in humans. The mutations investigated were the point mutation GFI1BT174N, H181Y and R184P and the missense mutation Q287*. The surrogate models investigated are the MEG01 megakaryoblast cell line and induced pluripotent stem cells derived from a patient carrying the Q287* mutation induced to differentiate along the megakaryocyte lineage. The effects of the different mutations were profiled by single cell RNA seq. Loss-of-function studies were performed by treating the cells with an inhibitor of LSD1.

The most important take-home message of the data presented is that the Q287* mutation appears to encode a dominant negative protein because it induces a molecular signature similar to that induced by the LSD1 inhibitor. This conclusion is clinically important because the phenotype expressed by patients carrying the Q287* mutation may predict the long-term effects of LSD1 treatment in patients with essential thrombocythemia and other myeloproliferative disorders.

1) Although the power conferred to the paper by single cell RNAseq analyses is greatly appreciated, the interpretation of the findings is greatly affected by the paucity of controls. The megakaryoblastic MEG-01 cell line was isolated long time ago (in 1983) from the marrow of a 55-year-old, male patient with Chronic Myelogenous Leukemia. At that time, this cell line contained the Ph chromosome and was hyper-diploid with a modal number of 56-58 chromosomes. The levels of endogenous GFI1b and LSD1 expressed by the cell line at baseline and after transfection should be presented. In addition, some type of assurance that the infections achieve comparable levels of expression of the different GFI1b variants should be presented. This assurance is important because the experiments with over-expression of GFI1b WT suggest that expression levels are important. Presently, it is difficult to assess whether the different phenotypes are sustained by mutation-specific effects or by technical issues which affected the expression levels of the various mutants.

We thank reviewer 2 for this comment and agree that being able to distinguish between technical and biological variation is important. We have added a new supplementary figure 1 that shows GFI1B and LSD1 expression levels in MEG01 samples transduced with GFI1B constructs or GFP and added it to the results section of the manuscript (page 9, ll. 175-179). We also included the fold-changes between GFP-transduced MEG01 cells and MEG01 cells transduced with the GFI1B constructs in the figure legend and text (page 9, l. 177). All

GFI1B overexpression samples were within 1-fold of each other and globally showed a 2-3 fold induction compared to empty vector. LSD1 expression remained unchanged when comparing GFP and GFI1B transduced cells.

2) Figure 1. The principal component analyses shown in this Figure is confusing. The cluster of the cells transfected with an empty vector (GFP) is separated by that of the cells transfected with GFI1B-q287* and with GFI1B WT. Moreover the GFI1B-Q287* cells resemble more closely GFI1B-T174N cells than WT cells. In any case, the variance is very low, indicating that the cells are overall very similar.

We appreciate that this is brought to our attention. We have adapted the results section (page 9, ll. 179-182) to highlight the similarities between replicates and differences between samples. We also mentioned that the variance is low for this PCA.

3) Figure 2. Again the module of the GFP cells is very different from that of the GFI1B WT, indicating that dosage is important.

Indeed, dosage of GFI1B is important. We have added a section to the results that highlights this importance (page 9, ll. 187-191).

4) Figure 1. The strategy to use mitochondrial genes and cell cycle genes to clean possible confounding factors is unclear given the number of reports indicating that mitochondrial and cell cycle genes are part of the megakaryocyte differentiation signature. Please clarify the rationale supporting this strategy.

We thank the reviewer for this comment and assume that this is related to Supplemental Figure 1 in the original version (now Supplemental Figure 2 in the updated version). Filtering for mitochondrial genes is a key quality control step in single-cell analysis. It removes cells that got damaged during sample preparation and subsequently lost their cytoplasmic RNA. Therefore, these should be omitted. We are aware of the fact that mitochondrial and cell cycle genes are important for MK differentiation. Therefore, we used a filtering strategy that allows us to specifically remove these damaged cells/droplets, while keeping cells that contain a high mitochondrial percentage but are still intact. We achieved this by not setting a hard mitochondrial gene threshold, which is the current standard method. Instead, we clustered the cells and looked for clusters that are characterized by high mitochondrial gene percentages. We then searched for megakaryocyte marker genes in those clusters. If we did not find any, we removed these clusters, otherwise they were kept for further analysis. We recognize that we did not explain this properly in the material and methods and adapted it to explain our rationale more clearly (page 7, ll. 127-131). Additionally, cell cycle is a confounding factor in our single cell analysis. We adapted the material and methods to better reflect our rationale (page 7, ll. 134-137).

5) Supplementary Figure 2 is confusing because the pie charts of the clusters in D do not recapitulate the heat maps in C. The pie chart of cluster 0 indicate a high percentage of megakaryocyte genes which are not present in its heat map. Cluster 0 and 8 appear very similar and it is confusing whether they represent progenitor or endothelial cells. What the word progenitor means, hematopoietic or endothelial progenitor cells? Cluster 4 appears to have a megakaryocyte rather than an Ery signature, as indicated.

We thank the reviewer for bringing this to our attention. To clarify the concern of the reviewer we wish to point out that cluster 0 and 8 look very similar when annotating them using the Descartes dataset. We annotated the cells using two independent methods, scGSEA and Azimuth. Combining the annotation for cluster 0 and 8 from panel C and D, we decided to name them myeloid progenitors (cluster 8) and megakaryocyte progenitors (cluster 0). Cluster 4 has an elevated signature for megakaryocytes but also erythroblasts and HSCs, albeit lower than the megakaryocyte signature. In combination with the Azimuth annotation in panel D it implies that megakaryocytes are the dominant cell type in this cluster, therefore we named it megakaryocyte progenitors. We expanded the material and methods (page 7, ll. 140-145) and figure description (Supplementary Material document, page 5, ll. 33-37) to reflect our reasoning and added a label to Supplementary Figure 3 panels C and D to highlight that these annotations were obtained using two independent tools.

6) Page 9: "GF11BQ287* fails to repress innate immunity genes in megakaryoblast-like cells." Recent data indicate that megakaryocytes represent four different sub-populations each one with a different function (Pariser et al. JCI 2021;131(1),e137377; Yeung et al Blood Adv 2020;4(24),6204-17; Wang et al. Cell Stem Cell 2021; 28(3),535-549.e8; Sun et al. Blood 2021;138(14),1211-24, among others). One of these populations has the signature of immune cells and is involved in the host response toward opportunistic infections in the lung. The expression signature of the different megakaryocyte subpopulations is published. It seems here a missed opportunity that the expression data presented in this manuscript are not compared with the signatures of the megakaryocyte subpopulations already published to assess whether the myeloid effects of these genes are exerted on "myeloid cells" or on "megakaryocyte subpopulations with myeloid functions".

We thank the reviewer to bring this to our attention as well. We have investigated the immunological signature found in the mentioned papers. Generally, we do not see that the immunological signatures reported in any of the studies visually correlated with a particular sample in our dataset (Ref.Fig1-3). There are several possible reasons for that: We are investigating a MEP cell line while the mentioned papers are looking into megakaryocytes isolated from mouse lungs or human yolk sac/foetal liver. The authors use single-cell RNA sequencing while we use bulk RNA sequencing for this dataset. This enabled them to look into a specific subpopulation which contains this signature. Our bulk data does not have the resolution to pick up any small subpopulations.

We have also investigated these signatures in our single-cell dataset. Megakaryocytes described in this manuscript do not show expression for these immunological signatures, while myeloid cells do. However, myeloid cells in our dataset do not show expression of megakaryocytes markers (Ref.Fig4). Immunological megakaryocytes reported in the above mentioned studies did express classical human/mouse MK markers. Based on the lack of finding an immunological signature in our datasets we decided to not alter the text.

Ref.Fig1: Expression of the Immunological gene signature found by Pariser et al. (Pariser al. JCI 2021;131(1),e137377) in our MEG01 dataset. There is no visual correlation between this gene signature and the samples

Ref.Fig2: Expression of the Immunological gene signature found by Yeung et al. (Yeung et al Blood Adv 2020;4(24),6204-17) in our MEG01 dataset. Yeung et al. identified immunological MK cluster in adult lungs from mouse. The signature does not show a visual correlation with a MEG01 sample from our dataset.

Ref.Fig3: Expression of the Immunological gene signature found by Wang et al. (Wang et al. Cell Stem Cell 2021; 28(3),535-549.e8) in our MEG01 dataset. They found a subpopulation of in-vitro derived megakaryocytes that express this immunological gene signature. Our MEG01 cells do not show a visual correlation with this signature.

7) Page 10, line 184. What was the rationale to study iPSC carrying the GF1BQ287* mutation in the context of this study? These experiments should include iPSC carrying the WT mutation as control.

We thank the reviewer for this comment. We have included an introductory sentence to this section and restructured the following sentence to better reflect our rationale for choosing to investigate GF1B^{Q287} during megakaryopoiesis (page 10, l. 206). We also restructured the next sentence to clear up any confusion related to our comparison between GF1B^{Q287*} and wild type iPSCs (page 10, ll. 209-213).*

8) Page 11, lines 210-211. The sophistication of the method used for GFP-painting of the endogenous LSD1 gene is greatly appreciated. It is unclear, however, what was the rationale for using the erythroleukemic K562 cell line for these experiments. As the megakaryoblastic MEG-01 cell line, the K562 cell line harbor the Ph chromosome but acquires megakaryocytic features only when treated with PMA. Results on PMA induced K562 cells would have been more relevant but are not presented. What is the copy number of the LSD1 gene in K562 and MEG-01 cells? Was on the basis of this copy number that determined the cell line to be used in these experiments. In any case, since LSD1 inhibitors are not considered for treatment in chronic myelogenous leukemia but in myeloproliferative disorders, the use of the HEL cell line which carries the JAK2V617F mutation would have provided more clinically relevant information.

The mass spec experiment was performed to independently confirm LSD1 interacting proteins and agree that HEL cell lines would provide interesting information in the context of myeloproliferative disorders. However, Yamamoto et al (Yamamoto et al., Oncotarget. 2018 Apr 20; 9(30): 21007–21021.) have investigated the endogenous interactome of LSD1 in HEL cells.

We used K562 cells to confirm LSD1 interaction partners because they can be readily CRISpaigned and we had them at our disposal. K562 and HEL cells both have four copies of the LSD1 gene, while MEG01 has two. Their LSD1 expression levels are similar according to DepMap and all express GFI1B (Ref.Fig 5). We have adapted the introductory sentence of this result section to better reflect the data that has been published and our rational in choosing K562 (page 11, ll. 237-238).

Ref.Fig 5: Expression of LSD1 and GFI1B in K562, MEG01, and HEL cells. Data was downloaded from DepMap (accessed 18.12.2023).

9) A diagram summarizing the location of the mutations analyzed, the trait expressed by the patients and the expression signature induced by their over-expression would be greatly beneficial. Ideally, this diagram should also indicate whether the patients carrying these mutations have defective innate immune responses.

We thank the reviewer for this suggestion, a diagram summarizing the patient data for each mutation has been added to the manuscript as Figure 1 and added to the results section (page 9, ll. 174-175). To our knowledge, individuals with GFI1B variants (at least the ones that we put forward here) do not have defective innate immune responses. Therefore, we decided to not include that in the Table.

10) The title is too restrictive and does not give sufficient credit to the novelty of the message provided by the paper.

We agree that the title is restrictive. At the same time, we feel that the title covers precisely the novelty of our findings without too much speculation. We have rewritten the title to make it active and put the emphasis on GFI1B and LSD1.

11) The conclusions are too dry and to not provide sufficient credit of the reasons why

the study is important.

We appreciate the reviewers' feedback on this. We feel that the findings of this manuscript are relevant as our data reveal that GFI1B and LSD1 have a dual function during megakaryopoiesis by allowing the expression of megakaryocyte programs (which is known) and at the same time suppress myeloid programs which was unknown until now. Additionally, we show that the myeloid master regulator SPI1 is regulated by GFI1B and LSD1, next to ETV5. Furthermore, our data shows that different GFI1B variants have distinct effects on the transcriptome in our in vitro model. The variant causing a severe bleeding phenotype is unique, in that it fails to repress myeloid programs. Other GFI1B variants did not result in the activation of these programs. To our knowledge, the effect of GFI1b variants on the transcriptome have not been investigated in this much detail. We have rewritten parts of the discussion to highlight the novelty of our findings.

Minor:

- Page 3, lines 48-50 and Ref. 14. Ref. 14 is a non-appropriate self-quotation. It is a review on the role played by LSD1 in AML and does not discuss the use of Bomedemstat to treat thrombocytosis in patients with myeloproliferative disorders, as stated by this paragraph.

We agree and removed the citation.

- The legend of all the Figures and Tables included in the supplementary should specify if the data refer to the MEG-01 cell line or iPSC derived cells.

We thank the reviewer for this suggestion. We added MEG01 and iPSC to the figure titles of Figures 5 and 6 and Supplementary Figures 1, 2, and 3 for clarity.

Reviewer #3 (Remarks to the Author):

The manuscript presents genomic data from cells transduced with expression vectors for GFI1B mutants found in human patients with thrombocytopenia and data sets from human ISPCs from patients with a GFI1B mutation. The data are well presented and support the conclusions of the authors.

The manuscript does not have any technical or conceptual flaws and the the conclusions are original. Data that were generated in murine models with Gfi1b mutant alleles to address thrombocytopenia have been published and the results could be discussed in relation to the insight gained here, if there is space.

The results presented are most certainly of immediate relevance for people in this field and even for a broader audience. The most compelling feature of this article is that fact that primary human stem cells have been used and patient derived mutants have been studied. Also hat convergence of results with mutant GFI1B forms and the LSD1 inhibitor nicely corroborates mechanistic models for human patients that have been suggested previously in cell lines or mouse models.

No additional experiments are needed, but the following point could be addressed by text to make a few points clearer:

- The authors use an LSD1 inhibitor, which would not only affect the function of GFI1B but also other transcription factors that associate with LSD1; which ones are those and could that affect the interpretation of the results?

We thank the reviewer for this very valid question. We have added a section to the result/discussion that discusses this. In short, we find regulons in our single-cell data that are only found after LSD1 inhibition and not GFI1B^{Q287}. These are likely effects of the LSD1 inhibitor independently of GFI1B (page 12f, ll. 270-278).*

- It has been shown that LSD1 inhibitors lead to the dissociation of LSD1 and GFI1B and to the eviction of both from the chromatin. Is it expected that this happen here as well with the specific inhibitor used here and the GFI1B mutant forms?

The reviewer states a very interesting, mechanistic question. Based on our data we cannot answer this question conclusively but our lab data shows that LSD1 and GFI1B cannot interact anymore in the presence of the GSK-LSD1 inhibitor. We expect that this impacts GFI1B-LSD1 retention on the chromatin.

Also, the GFI1B^{Q287} variant introduces a premature stop-codon in the DNA binding domain of GFI1B. We expect that this yields a protein that has impaired binding to the DNA. The other three variants are in the first zinc-finger, which is not necessary for DNA binding of*

GFI1B. As such, we expect the DNA binding capabilities of these variants to be more similar to wild type GFI1B.

- Also, does LSD1 act as an enzyme or a scaffold to recruit other regulators in this context? The authors mention this shortly, but could expand on that issue if there is space.

We thank the reviewer for this comment. The role of LSD1 as a scaffolding protein or as a demethylase is currently under a lot of scrutiny. Recent publications claim that the loss of the scaffolding function after LSD1 inhibition are responsible for downstream activation of transcription factors. This matches the mechanistic patterns that we see in our data. We think that the scaffolding function is more important during megakaryopoiesis and that the loss of this drives the phenotype. However, we have no data to prove that methylation patterns are not affected by LSD1 inhibition in our model system. We have added this to the end of the discussion (page 12f, ll. 271-278).

- The wild type endogenous GFI1B is still expressed in all the systems that the authors use: is the model then that the mutant forms act in a dominant negative manner? Or is the auto repression of GFI1B affected?

We thank the reviewer for this question. We have shown that the GFI1B^{Q287} functions acts in a dominant-negative manner (Monteferrario et al. N Engl J Med 2014; 370:245-253). It, as well as the other GFI1B variants, are found to be heterozygous in patients. Therefore, we think that all GFI1B variants investigated here are dominant-negative. In turn, this will affect the autoregulation of GFI1B. We added this to the introduction to highlight that the mutations are dominant-negative (page 4, ll. 64-65).*

- The introduction should mention that GFI1B has a paralogue (GFI1) and its role in myeloid cells. Also, that GFI1 is most certainly expressed in the stem cells that were used to generate megakaryocytes.

The reviewer raises a valid point. We have adapted our introduction to cover the function of GFI1 and GFI1B in hematopoietic stem cells and their role in myeloid and megakaryocyte differentiation, respectively (page 3, ll. 38-52).

REVIEWERS' COMMENTS:

Reviewer #2 (Remarks to the Author):

Thanks for your accurate revision of the manuscript. All my comments are appropriately addressed and I do not have any further comment.

Reviewer #3 (Remarks to the Author):

The authors have answered the reviewer comments and their revisions have addressed my comments in a satisfactory manner.